# Advanced Magnetic Resonance Imaging for Early Diagnosis and Monitoring of Movement Disorders

**DOI:** 10.3390/brainsci15010079

**Published:** 2025-01-16

**Authors:** Emmanuel Ortega-Robles, Benito de Celis Alonso, Jessica Cantillo-Negrete, Ruben I. Carino-Escobar, Oscar Arias-Carrión

**Affiliations:** 1Unidad de Trastornos del Movimiento y Sueño, Hospital General Dr. Manuel Gea González, Calzada de Tlalpan 4800, Mexico City 14080, Mexico; emmanuel.ortega@salud.gob.mx; 2Facultad de Ciencias Físico Matemáticas, Benemérita Universidad Autónoma de Puebla, Puebla 72570, Mexico; bdca@fcfm.buap.mx; 3Technological Research Subdirection, Instituto Nacional de Rehabilitación Luis Guillermo Ibarra Ibarra, Mexico City 14389, Mexico; jcantillo@inr.gob.mx; 4Division of Research in Clinical Neuroscience, Instituto Nacional de Rehabilitación Luis Guillermo Ibarra Ibarra, Mexico City 14389, Mexico; ricarino@inr.gob.mx

**Keywords:** magnetic resonance imaging, movement disorders, Parkinson’s disease, neurodegeneration biomarkers, neuroimaging, early diagnosis

## Abstract

Advanced magnetic resonance imaging (MRI) techniques are transforming the study of movement disorders by providing valuable insights into disease mechanisms. This narrative review presents a comprehensive overview of their applications in this field, offering an updated perspective on their potential for early diagnosis, disease monitoring, and therapeutic evaluation. Emerging MRI modalities such as neuromelanin-sensitive imaging, diffusion-weighted imaging, magnetization transfer imaging, and relaxometry provide sensitive biomarkers that can detect early microstructural degeneration, iron deposition, and connectivity disruptions in key regions like the substantia nigra. These techniques enable earlier and more accurate differentiation of movement disorders, including Parkinson’s disease, progressive supranuclear palsy, multiple system atrophy, corticobasal degeneration, Lewy body and frontotemporal dementia, Huntington’s disease, and dystonia. Furthermore, MRI provides objective metrics for tracking disease progression and assessing therapeutic efficacy, making it an indispensable tool in clinical trials. Despite these advances, the absence of standardized protocols limits their integration into routine clinical practice. Addressing this gap and incorporating these techniques more systematically could bring the field closer to leveraging advanced MRI for personalized treatment strategies, ultimately improving outcomes for individuals with movement disorders.

## 1. Introduction

Movement disorders, particularly Parkinson’s disease (PD), affect millions worldwide and remain among the most challenging neurological conditions to diagnose and manage. Traditional diagnostic methods, heavily reliant on clinical symptoms, often fail to detect early neurodegenerative changes, limiting opportunities for timely intervention [1,2]. Recent advancements in magnetic resonance imaging (MRI) have introduced new tools that enable detailed visualization of structural and functional brain changes. While conventional T_1_- and T_2_-weighted imaging techniques are widely used, they lack the sensitivity needed to detect subtle neurodegenerative alterations characteristic of early PD and related movement disorders.

Advanced MRI modalities, such as neuromelanin-sensitive imaging (NM-MRI), diffusion-weighted imaging (DWI), magnetization transfer imaging, and relaxometry, have emerged as powerful tools for the early detection of neurodegenerative processes, particularly within the substantia nigra. These techniques enhance the visualization of dopaminergic neuron loss and provide insights into iron deposition and disrupted neural networks, all of which are implicated in the progression of PD. Additionally, their utility extends beyond PD, aiding in the differentiation of atypical parkinsonisms such as progressive supranuclear palsy, multiple system atrophy, corticobasal degeneration, and Lewy body dementia [3].

Despite these technological advancements, significant challenges persist. The integration of advanced MRI into routine clinical practice remains in its early stages, and standardization across institutions is essential to fully realize their diagnostic and prognostic potential [4]. Additionally, the incorporation of artificial intelligence holds promise for enhancing the analysis of complex imaging datasets, potentially enabling the development of predictive models that support personalized therapeutic strategies [5,6].

This review begins with a theoretical overview of MRI techniques applied to the study of movement disorders. It then critically examines the current landscape of advanced MRI methodologies in this field, emphasizing their contributions to early diagnosis, tracking disease progression, and guiding the design of clinical trials.

By addressing these challenges, we aim to highlight the transformative potential of MRI in advancing personalized medicine for movement disorders.

## 2. Magnetic Resonance Imaging Techniques

From a physical standpoint, MRI is a technology that utilizes powerful magnetic fields combined with radiofrequency pulses (RF) to create detailed images of human anatomy. It induces the H^+^ protons in water molecules to resonate at a specific frequency (Larmor’s frequency) dictated by a primary external magnetic field (B_0_). Using RF pulses in conjunction with magnetic gradient fields, MRI leverages the T_1_, T_2_, and T_2_* relaxation times to obtain anatomical images [7]. The advantages of this technology—such as not relying on ionizing radiation, providing high spatial resolution images and excellent contrast between tissues, and offering various contrast mechanisms—make it an ideal tool for diagnosing and monitoring the progression of a wide range of movement disorders [8].

In the following sections, we provide an overview of the main MRI applications, categorizing techniques into qualitative and quantitative as defined by Mills et al. [9]. Qualitative MRI techniques perform diagnoses or research based on the visual interpretation of image changes rather than specific quantitative measurements. Quantitative MRI, on the other hand, focuses on measuring and interpreting specific parameters derived from the MRI data. Table 1 provides a concise summary of these techniques.

### 2.1. Qualitative MRI

#### 2.1.1. Structural Image Analysis Techniques—Traditional Imaging with Proton Density, T_1_, T_2_, and T_2_* Contrasts

Basic MRI techniques are primarily used to obtain detailed structural or anatomical images. These techniques measure the relaxation properties of water molecules, specifically how quickly their nuclear spins return to equilibrium after being disturbed by an RF pulse. The key relaxation parameters include T_1_, T_2_, T_2_*, and proton density [7].

T_1_, or spin–lattice relaxation time, reflects the time required for spins to realign along the *Z*-axis after the absorption of RF energy. It provides a measure of how the system dissipates the excess energy introduced by the RF pulse into its surrounding environment. T_2_, also known as spin–spin relaxation time, measures the loss of coherence in the precession of spins in the XY plane. This dephasing occurs due to interactions between neighboring water molecules, which create variations in the local magnetic field. T_2_*, a related parameter, accounts for additional dephasing caused by magnetic field inhomogeneities from external sources, such as iron deposits or tissue boundaries, rather than intrinsic molecular interactions. Proton-density-weighted imaging (PDWI) is a basic measurement of the water content in tissues, offering insights into their composition [10].

While conventional T_1_, T_2_, and proton density imaging methods have traditionally been used in neuroimaging, their effectiveness in diagnosing or monitoring movement disorders has diminished in recent years. These methods have largely been supplanted by more advanced MRI techniques, including functional MRI (fMRI), diffusion-weighted imaging (DWI), diffusion tensor imaging (DTI), voxel-based morphometry (VBM), and cortical thickness analysis. These advanced techniques are now primarily focused on identifying biomarkers for the diagnosis and monitoring of movement disorders, offering greater sensitivity and specificity in detecting the subtle pathological changes associated with these conditions.

#### 2.1.2. Adiabatic Techniques

Adiabatic techniques provide measurements of relaxation times that differ from those obtained with traditional methods, such as T_1_ and T_2_ relaxation times. Examples of these parameters include T_1ρ_ and T_2ρ_ [11,12]. These techniques utilize adiabatic pulses, a specialized form of RF pulse designed to refocus, invert, or excite molecular spins, even in regions with significant magnetic field inhomogeneities. This capability makes adiabatic techniques particularly advantageous for addressing challenges posed by magnetic susceptibility variations.

T_2ρ_ is highly sensitive to regions with elevated iron content, making it a valuable tool for detecting iron accumulation that may be associated with neuronal dysfunction. In contrast, T_1ρ_ focuses on water–protein interactions, providing insights into cellular integrity and potential damage to or loss of brain cells.

#### 2.1.3. Voxel-Based Morphometry

Voxel-based morphometry (VBM) is a technique employed to assess tissue composition within the brain, specifically quantifying the relative amounts of grey and white matter in each voxel. Unlike methods that measure brain volume, VBM focuses on characterizing the tissue composition of brain regions. This technique relies on highly detailed anatomical images, typically acquired using T_1_- and T_2_-weighted sequences, to analyze the entire brain. By comparing brain structures across different patient cohorts, VBM enables the derivation of meaningful conclusions regarding structural differences and variations. For a comprehensive review of the technique and its methodologies, we refer the reader to [13].

#### 2.1.4. Susceptibility Weighted Imaging and T_2_* Measurements

T_2_* and susceptibility-weighted imaging (SWI) are techniques that quantify susceptibility effects between tissues, with a primary focus on assessing iron deposition. T_2_* imaging employs gradient echo sequences and relies on variations in T_2_* relaxation times caused by elements that perturb the local magnetic field, such as iron. In contrast, SWI provides a more comprehensive analysis by integrating both magnitude and filtered phase images. The phase component, derived from a gradient recalled echo sequence, captures information about regions with significant susceptibility effects, including veins, iron deposits, and hemorrhages. These regions appear darker when overlaid onto the magnitude image, enhancing the visualization of susceptibility-related features [14].

#### 2.1.5. Neuromelanin-Sensitive Imaging

Neuromelanin-sensitive MRI (NM-MRI) is a specialized technique designed to detect neuromelanin, a pigment present in brain regions such as the substantia nigra (SN) and locus coeruleus (LC), which are commonly affected in neurodegenerative conditions like PD. Neuromelanin is synthesized through the oxidation of catecholamines in these regions and increases linearly with natural aging. In PD, the pigment is lost as affected neurons degenerate. NM-MRI exploits the fact that neuromelanin binds metals such as iron or copper, resulting in hyperintense regions on MRI scans. Unlike susceptibility-based techniques, NM-MRI integrates the relaxometry of T_1_ and T_2_ with quantitative magnetization transfer (QMT) to assess neuromelanin content. The most significant signal changes are observed in the T_1_ contrast, which also mediates the QMT measurements through the T_1_ effect [15].

#### 2.1.6. Functional MRI

Functional MRI (fMRI) comprises a group of techniques designed to measure hemodynamic changes associated with central nervous system (CNS) activity. Although the precise mechanisms underlying the coupling of neuronal activity and hemodynamic response remain a topic of ongoing research, these changes can be detected by MRI scanners due to the distinct magnetic properties of oxygenated and deoxygenated blood, a phenomenon known as the blood oxygen level-dependent (BOLD) contrast [16].

The two primary techniques within fMRI are task-based fMRI (tbfMRI) and resting-state fMRI (rsfMRI). In tbfMRI, participants perform specific tasks within the scanner, such as moving a hand or viewing visual stimuli. These tasks elicit increased oxygenated blood flow to brain regions processing the task-related information, resulting in elevated MRI signal intensity in those regions, which is then detected and highlighted. Imaging is typically performed using fast T_2_* sequences, making tbfMRI, in part, a susceptibility-based measurement [17].

In contrast, rsfMRI involves acquiring continuous T_2_* images over a short period (typically 2–10 min) while participants remain at rest without performing specific tasks. The BOLD time series from different regions of interest (ROIs) are extracted and filtered to focus on low-frequency fluctuations (0.001–0.1 Hz). These low-frequency oscillations reveal functional networks characterized by correlated activity among brain regions, even when spatially distant. Examples of such networks include the auditory, visual, motor, and executive control systems [17].

Both tbfMRI and rsfMRI provide insights into brain activation patterns during tasks and the functional connectivity (FC) between brain regions. FC reflects the coordinated activity of brain areas and offers critical information about the organization of brain networks and their potential disruptions in neurological and psychiatric disorders [18].

#### 2.1.7. Diffusion-Weighted Imaging and Diffusion Tensor Imaging

Diffusion-weighted imaging (DWI) is a technique used to study the movement of water molecules within tissues. While water molecules typically exhibit random thermal motion (isotropic diffusion), their movement within a voxel of human tissue is shaped by the tissue’s structural characteristics, cellular architecture, and composition, leading to more directional movement (anisotropic diffusion).

To perform DWI, standard T_1_ and T_2_ imaging are initially acquired. A pair of opposing gradients is then applied to the tissue. The changes in T_1_ and T_2_ signals observed during this process reflect the displacement of water molecules that leave and enter the voxel due to diffusion. This provides insights into how water moves through brain tissues, offering information about cellular integrity and function [19].

Diffusion tensor imaging (DTI), a variant of DWI, specifically measures water diffusion within neurons, where a preferred motion along axons is observed. DTI enables the visualization of brain tracts, representing the physical connections between regions of interest. This structural connectivity contrasts with functional connectivity, as it reflects the anatomical pathways linking brain areas [20].

Both DWI and DTI provide valuable information through parameters such as the apparent diffusion coefficient (ADC), fractional anisotropy (FA), mean (MD), axial diffusivity (AD), or radial diffusivity (RD). The ADC, derived from DWI images acquired at varying gradient strengths, measures the magnitude of diffusion in mm^2^/s. In ADC maps, regions with high diffusion appear bright, while those with low diffusion appear dark, reflecting the influence of cellular microstructure. This parameter is particularly useful for distinguishing pathological from non-pathological tissues. FA quantifies the degree of anisotropy, with values close to 1 indicating a strong preferred direction of diffusion (e.g., in well-organized axonal tracts) and values close to 0 indicating isotropic diffusion. FA provides insights into axonal diameter, fiber density, and myelination. MD represents the overall magnitude of water diffusion, averaged across all directions, and is often used to assess tissue integrity, with increased values indicating degeneration or oedema. AD measures water diffusion along the primary axis of the diffusion tensor and is typically associated with axonal integrity, with reductions suggesting axonal injury. In contrast, RD quantifies diffusion perpendicular to the principal axis and is linked to myelin integrity, with elevated values often indicating demyelination. Together, these parameters aid in understanding structural changes in neurological conditions [20].

#### 2.1.8. Magnetic Resonance Elastography

Magnetic resonance elastography (MRE) is a technique designed to measure the elastic and compressible properties of tissues, particularly in soft tissues such as the liver, breast, and prostate. It integrates anatomical MRI imaging, typically acquired using T_1_-weighted sequences, with a shear stress image, referred to as the in-phase image. To generate the shear waves, pressure waves are mechanically induced in the tissue of interest using an external transducer. These waves are then encoded using a motion-sensitizing gradient during the acquisition of an echo-planar imaging (EPI) sequence, producing the in-phase image. When combined with the anatomical MRI image, this process generates an elastogram, which allows the assessment of various MRE parameters, providing insights into the mechanical properties of the tissue [21].

### 2.2. Quantitative MRI

Thus far, the techniques discussed have primarily yielded qualitative results in the form of images, which are typically compared visually [22]. The subsequent techniques described herein, however, represent quantitative MRI approaches that provide numerical measurements of specific parameters. These quantitative values enable direct comparison and analysis, offering a more precise and objective assessment of tissue properties and their variations.

#### 2.2.1. Morphometric Measurements (Cortical Thickness and Brain Volumes)

Cortical thickness is a morphometric measurement that quantifies the thickness of the brain’s outer layer, known as the cortex [23]. This layer contains most of the gray matter and plays a critical role in numerous brain functions. Cortical thickness is known to be influenced by various disorders and illnesses and is expressed as a length value in millimeters. Brain volume measurement, another form of morphometric analysis, quantifies the volume occupied by specific brain structures, typically reported in cubic millimeters or as the number of voxels.

For both cortical thickness and brain volume measurements, researchers require high-quality T_1_- and T_2_-weighted images, preferably with resolutions of 1 mm^3^ or finer, to ensure accurate and reliable results. Some studies have recommended the use of magnetization-prepared rapid gradient-echo (MPRAGE) sequences to correct for B_1_ field inhomogeneities, thereby enhancing the precision of the measurements [24].

#### 2.2.2. Quantitative Magnetization Transfer

Quantitative magnetization transfer (QMT) is an MRI technique that quantifies the exchange of spin properties, such as coherence or polarization, between molecules within a tissue or sample. The most studied scenario involves the exchange between free and bound water molecules in tissue. QMT measures the extent to which magnetization transfer occurs between these two pools. To assess these effects, an RF saturation pulse is applied at a frequency selectively absorbed by bound water, thereby saturating its signal while leaving the free water signal unaffected. A T_1_-weighted image acquired under these conditions reflects primarily the T_1_ relaxation of free water, unperturbed by the attenuation of bound water. Several factors influence QMT measurements, including water content, membrane fluidity, and heavy metal concentrations [7].

#### 2.2.3. Perfusion MRI

Perfusion MRI (pMRI) techniques encompass arterial spin labeling (ASL), dynamic susceptibility contrast (DSC), and dynamic-contrast-enhanced (DCE) imaging, which are used to assess the vasculature of tissues and quantify the dynamics of fluid perfusion, providing insights into tissue health. From these techniques, quantitative parameters such as cerebral blood volume (CBV), cerebral blood flow (CBF), and time to peak (TTP) can be derived.

In DSC and DCE imaging, a gadolinium-based contrast agent is injected, and fast T_2_*- and T_1_-weighted susceptibility-based sequences are employed to monitor the perfusion of the contrast agent into the tissue of interest. These techniques enable the assessment of vascular parameters and tissue perfusion. In contrast, ASL utilizes a non-invasive approach by selectively saturating the signal of arterial blood before it enters the brain. Subsequent imaging captures the perfusion-related signal changes, which are derived from the difference between pre- and post-saturation images. This technique is particularly valuable for evaluating ischemic conditions, vascular abnormalities, and neurodegenerative diseases [25].

#### 2.2.4. Magnetic Resonance Spectroscopy

Magnetic resonance spectroscopy (MRS) is an MRI-based spectroscopic technique that quantifies the levels of various chemicals or metabolites within the body, providing insights into cellular processes. This technique generates a chemical spectrum, which serves as a histogram of the elements present in a tissue sample, indicating their relative abundances. Unlike conventional MRI, MRS does not produce images, as it does not require spatial encoding gradients. Instead, it selectively excites the nuclei of elements such as hydrogen (H^+^), carbon (C), or sulfur (S), which are integral to specific molecules of interest.

Among the key metabolites measured by MRS are N-acetyl aspartate (NAA), creatine (Cre), and choline (Cho). These metabolites serve as biomarkers for various physiological and pathological processes. NAA is often used as a marker of neuronal integrity and function. A decrease in NAA levels is indicative of neuronal damage, loss, or dysfunction, making it a valuable indicator in neurodegenerative diseases. Choline is primarily associated with cell membrane turnover and phospholipid metabolism. Elevated levels of Cho are often observed in conditions involving increased cellular turnover, such as inflammation, demyelination, or malignancies. Creatine serves as a reference metabolite in MRS studies, as it is relatively stable across different physiological conditions. Cre is involved in energy metabolism and is present in both neurons and glial cells. Its concentration is often used to normalize metabolite levels, allowing more accurate comparisons between different regions or subjects [26].

### 2.3. Other Magnetic Resonance Techniques

Other MR techniques and applications hold potential for enhancing the diagnosis and monitoring of movement disorders. Chemical shift imaging, for instance, enables the quantification of water and fat content within a voxel through in-phase and out-of-phase measurements. This technique provides valuable information on tissue composition and is widely used for the diagnosis of benign and malignant tumors, particularly in body and musculoskeletal MRI. However, its application in movement disorders is limited, with most studies focusing on animal models.

In recent years, the integration of artificial intelligence (AI), including subdisciplines such as machine learning, natural language processing, and neural networks, has revolutionized medical diagnostics and monitoring. AI’s applicability to MRI is extensive, offering advancements in disease diagnosis and monitoring, image artifact correction, segmentation, clinical workflow optimization, and biomarker discovery [5,6]. These processes can be automated, reducing the need for manual intervention. While AI holds significant promise for the field of movement disorders, it is beyond the scope of this work and warrants a comprehensive review in its own right. Consequently, further discussion of AI applications in this context will not be included here.

## 3. Methods

A comprehensive literature search was conducted up to April 2024 using the databases PubMed, Scopus, Web of Science, and Google Scholar. The search strategy included combinations of keywords related to movement disorders (e.g., “Movement disorders”, “Parkinson’s disease”, “progressive supranuclear palsy”, “multiple system atrophy”, “corticobasal degeneration”, “Lewy body dementia”, “frontotemporal dementia”, “Huntington’s disease”, “dystonia”) and MRI techniques (e.g., “magnetic resonance imaging”, “proton density-weighted imaging”, “neuromelanin-sensitive imaging”, “voxel-based morphometry”, “susceptibility-weighted imaging”, “fMRI”, “diffusion-weighted imaging”, “diffusion-tensor imaging”, “magnetic resonance elastography”, “magnetization transfer imaging”, “relaxometry”, “Perfusion MRI”, “magnetic resonance spectroscopy”). The search was restricted to articles written in English. Eligibility was determined through a two-step screening process performed by two authors (O.A.-C. and E.O.-R.). First, titles and abstracts of identified articles were screened. Second, full texts of potentially relevant studies were reviewed, and final inclusion was determined through discussion and consensus. As this is a narrative review, article selection was based on the authors’ discretion, and no formal systematic review guidelines or methodologies were applied. Included studies comprised case reports, observational studies, controlled trials, and systematic reviews that specifically investigated MRI techniques for the diagnosis, monitoring, or evaluation of the aforementioned movement disorders.

The studies included in Tables 2–4 were selected at the authors’ discretion based on their representativeness of the main findings and their significance as key references for the use of MRI techniques in diagnosing and monitoring movement disorders. The selection process relied on consensus among the authors and did not follow a systematic methodology.

## 4. MRI in Movement Disorders

Several reviews have explored the application of MRI techniques in PD [27] and, to a lesser extent, in some atypical parkinsonisms [28]. However, a comprehensive review encompassing the use of MRI across a broader range of movement disorders remains unavailable. The following sections discuss the use of various MRI techniques in PD, progressive supranuclear palsy (PSP), multiple system atrophy (MSA), corticobasal degeneration (CBD), Lewy body dementia (LBD), frontotemporal dementia (FTD), Huntington’s disease (HD), and dystonia (DTN). Table 2 provides a summary of key references on MRI for diagnosing and monitoring PD. Similarly, Table 3 summarizes MRI studies for disorders related to PD, including PSP, MSA, CBD, LBD, and FTD, while Table 4 focuses on HD and DTN.

### 4.1. Parkinson’s Disease

The use of conventional MRI measurements, such as T_1_, T_2_, and T_2_*, has been increasingly supplanted by advanced techniques such as relaxometry, fMRI, and DWI, which have proven valuable for identifying biomarkers of neurodegeneration, including substantia nigra pathology in PD [29]. However, qualitative studies of T_2_ and T_2_* relaxation times in PD patients have consistently demonstrated shorter relaxation times and increased relaxation rates (R_2_ and R_2_*) in the SN compared to healthy controls. These alterations are thought to reflect elevated iron content, with R_2_* changes being particularly pronounced in the lateral SN and correlating with motor scores on the Unified Parkinson’s Disease Rating Scale (UPDRS) [30,31]. Additionally, T_2_* hypointense signals are more prominent in the SN pars compacta (SNpc) of PD patients [32]. Histological findings corroborate these observations, revealing preferential cell loss in the lateral SN, which corresponds to its sensorimotor region. T_1_ measurements in PD patients have identified hypointense lacunes in the SN and locus coeruleus, while T_2_ measurements have highlighted hyperintense regions indicative of perivascular space enlargement in the semioval center [33]. Ratio imaging, a technique derived from two inversion-recovery sequences (one suppressing white matter and the other suppressing gray matter), has successfully distinguished PD patients from controls by revealing structural changes in the SN, including more pronounced thinning in the lateral segments compared to the medial segments, consistent with established SN pathology in PD [34]. Volumetric analyses have further demonstrated reduced striatal and hippocampal volumes in PD patients compared to controls, with progressive atrophy over time [35]. Proton density measurements have revealed decreased signal intensity in several brain regions, including the prefrontal subcortical area, pons, globus pallidus, and bilateral structures such as the splenium of the corpus callosum, caudate nucleus, putamen, thalamus, and mesencephalon. T_1_ relaxation times were increased in the prefrontal subcortical area, centrum semiovale, putamen, caudate nucleus, and mesencephalon, while T_2_ relaxation times were elevated in the pons and centrum semiovale. T_2_* relaxation times did not exhibit significant changes [36].

Advances in pulse sequences and relaxometry techniques, including the use of adiabatic pulses, have enhanced the detection of PD-related abnormalities. For instance, T_2ρ_ imaging has outperformed conventional T_2_ in identifying iron-related changes in the SN, while T_1ρ_ imaging has revealed neuronal loss in this region [11,12]. Mangia et al. [37] reported longer T_1ρ_ values in the amygdala and SN of PD patients, indicative of neuronal degeneration, whereas T_2ρ_ imaging demonstrated changes in the midbrain of patients with idiopathic rapid eye movement sleep behavior disorder (iRBD), suggesting altered motor function in this structure. Early voxel-based and region-of-interest studies reported no significant differences in SN volumes between PD patients and controls [38], but subsequent studies identified reduced brain volumes in the frontal cortex of PD patients [39]. Zeighami et al. [40] observed atrophy in the basal ganglia, midbrain, basal forebrain, and medial temporal lobe in PD patients, while VBM revealed reduced volumes in the anterior cingulate cortex [41]. Mak et al. [42] found reduced cortical thickness, particularly in the frontal cortex and temporal-parietal lobes, in newly diagnosed PD patients with and without cognitive impairment, along with atrophy of the nucleus accumbens. Reduced cortical thickness in the basal ganglia, sensorimotor, and frontal regions was strongly associated with poorer motor outcomes [43]. Structural connectivity to subcortical nuclei, sensorimotor, and frontal regions was also linked to better motor outcomes. SWI has identified reductions in T_1_ and T_2_* signals in the SN pars compacta and habenular nuclei of PD patients, along with increased subthalamic nucleus mean quantitative susceptibility mapping (QSM) values [44].

NM-MRI sequences have demonstrated a marked reduction in SNpc signal intensity in PD patients, reflecting the loss of neuromelanin-containing neurons [45]. This technique is particularly valuable for distinguishing PD from PSP and for monitoring disease progression [2]. Similar reductions in melanin signal have been observed in the locus coeruleus (LC), another key structure involved in PD [46,47]. Pavese et al. [48] highlighted the utility of melanin-sensitive imaging as a biomarker for differentiating PD patients from controls, though its efficacy in distinguishing PD from atypical parkinsonian syndromes remains limited. Biswas et al. [49] confirmed the utility of midbrain atrophy and putaminal hypointensity as biomarkers for distinguishing PD from both controls and parkinsonism patients. Reduced melanin signals have also been observed in the frontal lobe, hippocampus, basal ganglia, anterior cingulate cortex, superior temporal gyri, olfactory bulb, orbitofrontal cortex, ventrolateral prefrontal cortex, and occipitoparietal cortex [50].

Functional MRI studies have revealed changes in resting-state functional connectivity between PD patients and controls. For example, PD patients exhibit reduced connectivity between the posterior putamen and the inferior parietal cortex, while increased connectivity between the anterior putamen and the same region may represent a compensatory mechanism for motor deficits [30]. Functional connectivity changes have also been linked to parkinsonian tremor, with increased connectivity observed between the internal globus pallidus and putamen with the cerebellothalamic circuit in tremor-dominant PD patients. Dopamine depletion in the globus pallidus, but not the striatum, correlates with the severity of clinical tremor, suggesting that resting tremor may result from abnormal interactions between the basal ganglia and the cerebellothalamic circuit [51]. During movement initiation, PD patients show reduced connectivity between the putamen and key motor areas, including the primary motor cortex, premotor cortex, and supplementary motor area, compared to controls. Reduced connectivity during motor learning tasks is negatively correlated with clinical measures of motor function, indicating impaired neural coding and subcortical processing in PD [52]. Saeed et al. [50] reviewed altered functional connectivity biomarkers in PD, identifying changes in the default mode, salience, central executive, and sensorimotor networks, as well as in connections involving the basal ganglia, thalamocortical, cerebellothalamic, and cortical–subcortical sensorimotor circuits.

DWI and DTI studies have shown that mean diffusivity remains unchanged in the SN and basal ganglia of PD patients, both in region-of-interest and voxel-based analyses [53,54]. However, cerebellar axonal integrity is affected in PD, with the degree of impairment varying according to disease duration and symptomatology [55]. In contrast, increased diffusivity is consistently observed in PSP, particularly in the midbrain. Some studies have reported increased diffusivity in the striatum and thalamus of PD patients [56]. Reduced FA in the SN of early-stage PD patients (Hoehn and Yahr stages I–II) has been widely reported [57,58,59], though this finding has not been consistently replicated in smaller samples of early-stage patients [60,61]. Changes in R_2_* and FA in the SN have been shown to accurately distinguish PD patients from controls, with an overall accuracy of 95% using logistic regression models and receiver operating characteristic (ROC) curves [50]. DTI has also been used to study white matter connectivity in PD, revealing reduced connectivity between the SN and the ipsilateral putamen and thalamus, along with increased connectivity between the SN and the left caudate and globus pallidus. Diffusion-based automatic segmentation of SN subregions has identified generalized atrophy in both the SN pars reticulata and pars compacta in PD patients [62]. Shih et al. [63] reviewed studies showing reduced FA values in PD patients compared to controls in the nigrostriatal projections, thalamic subnuclei, corticospinal tract, and cingulum, among other regions. Mitchell et al. [64] highlighted the utility of DTI, SWI, neuromelanin imaging, and structural imaging as biomarkers for diagnosing PD and monitoring its progression.

MRE has shown marginal utility in studying brain and movement disorders, including PD, where it has demonstrated a negative correlation between disease severity and MRE parameters such as stiffness and viscosity [65,66,67]. QMT techniques have revealed a 15% reduction in SN volume in PD patients, with improved contrast and delineation of the SN. The magnetization transfer ratio (MTR) is significantly reduced not only in the SN but also in other basal ganglia structures, including the globus pallidus, putamen, and caudate nucleus [68,69]. Higher-field MRI scanners may further enhance the detection of SN pathology in PD. QMT imaging has also been used to study myelination, providing detailed images of deep brain structures by capturing the transfer of magnetization between tightly bound protons and free-floating protons in water [68]. Reduced macromolecular proton fraction, a QMT biomarker for myelin, has been observed in Huntington’s disease [70].

Perfusion techniques, including ASL and CBF measurements, have identified hypoperfusion in PD patients in regions such as the precuneus, occipital lobe, and sensorimotor area, indicative of neuronal loss and cerebrovascular dysfunction [71]. MRS has investigated metabolite ratios in the basal ganglia and cortex, with reduced NAA/Cho or NAA/Cre ratios observed in PD patients [72,73]. Due to the small size of the SN, few MRS studies have focused on this region in PD [74,75], though high-field MRI scanners (4T and higher) have improved the signal-to-noise ratio and spectral resolution, enabling more detailed studies of neurochemical metabolites in the SN. A study using 4T MRI detected 10 neurochemical metabolites in the SN of PD patients, including γ-aminobutyric acid (GABA), glutamate, and glutathione [76]. Comparisons between PD patients and controls have shown trends toward reduced glutathione and NAA in the anterior midcingulate cortex and increased Cho in the frontal white matter, with Cho being associated with inflammation and NAA with anti-inflammatory properties [77]. Reduced NAA/Cre ratios have been observed in the lateral nucleus, SN, temporoparietal cortex, posterior cingulate cortex, and pre-SMA [50]. MRS has also identified the thalamus as a critical region for assessing cognitive decline in PD, with reduced lactate and choline concentrations suggesting metabolic changes linked to disease progression [78].

**Table 2 brainsci-15-00079-t002:** Parkinson’s disease. A compilation of what the authors consider the key references on the use of MR techniques for diagnosing and monitoring PD, focusing on recent studies.

Study	MR Techniques	Main Findings
Sasaki et al., 2006 [45]	NM-MRI (T1-FSE)	↓ Signal in SN and LC.
Helmich et al., 2009 [51]	fMRI	↓ FC of posterior putamen and primary and secondary somatosensory cortex, IPC, insula, and cingulate motor area. ↑ FC of anterior putamen and IPC.
Lipp et al., 2013 [66]	MRE	↓ Viscosity in lentiform nucleus. Strong correlation between UPDRS-Part III and MRE parameters.
Ziegler et al., 2013 [79]	T_1_, T_2_, PDWI, FLAIR	↓ SNpc volume in early-PD (Hoehn and Yahr stage 1). ↓ SNpc volume, more pronounced in Hoehn and Yahr stages 2 or 3. ↓ Cholinergic basal forebrain volume in Hoehn and Yahr stages 2 or 3.
Langley et al., 2016 [59]	DWI, DTI, NM-MRI	↓ FA in SN (maximal in rostral and lateral). ↓ T_2_* in SN.
Saeed et al., 2020 [50]	T_1_, T_2_, T_2_*, R_2_*, PDWI, NM-MRI, FLAIR, SWI, DWI, DTI, fMRI, MRS	↓ Signal in frontal lobe, basal ganglia, hippocampus, anterior cingulate and superior temporal gyri, olfactory bulb, orbitofrontal, ventrolateral prefrontal, occipitoparietal, left cuneus, praecuneus, lingual gyrus, posterior cingulate, right perisylvian, and inferior temporal cortex. FC abnormalities in networks (default mode, salience, central executive, sensorimotor) and circuits (basal ganglia–thalamocortical, cortical–subcortical–sensorimotor, cerebellothalamic). ↓ FA in SN and olfactory bulb. ↑ MD in olfactory tract. ↓ NAA/Cre in LN, SN, temporoparietal and posterior cingulate cortex, and pre-SMA.
Alves et al., 2022 [33]	T_1_, T_2_, NM-MRI (T1-FSE), FLAIR, DWI	↓ Area and signal of NM-MRI in SN and LC. = Frequency of lacunes, micro-beds, infarcts or enlarged perivascular spaces.
Donahue et al., 2022 [77]	T_1_, MRS	↓ NAA associated with greater PVS volume in anterior middle cingulate cortex. ↑ Cho associated with greater PVS volume in frontal white matter.
Haghshomar et al., 2022 [55]	DWI, DTI	↓ FA, ↑ MD, ↑ RD, and ↑ AD in cerebellum.
Brown et al., 2023 [35]	T_1_, T_2_, DWI	↓ Volume in striatum. Faster decline in hippocampal volume, FSWM FA, and fornix FA.
Feng et al., 2023 [65]	MRE	↓ Shear modulus magnitude of global brain (mostly in frontal lobes and mesencephalic region).
Joshi et al., 2023 [71]	pMRI (ASL)	↓ CBF in basal ganglia (caudate nucleus, putamen, globus pallidus), frontoparietal network, precuneus, occipital lobe, sensorimotor area regions, visual network.
Marapin et al., 2023 [52]	fMRI	FC abnormalities in cerebellum, sensorimotor cortex, SMA, prefrontal cortex, thalamus, and insula. ↓ Local efficiency and ↑ global efficiency in the functional brain network. ↑ FC of sensorimotor network and cognitive control network. ↑ FC dentate nucleus and cerebellum. ↓ FC dentate nucleus and caudate.
Shih et al., 2023 [63]	DWI, DTI	↓ FA in corpus callosum, internal and external capsule, corona radiata, posterior thalamic radiation, sagittal stratum, cingulum, superior longitudinal fasciculus, STN. ↑ MD in internal and external capsule, corona radiata, corpus callosum, cingulum, fronto-occipital and longitudinal fasciculus.
Duan et al., 2024 [44]	T_1_, T_2_*, PDWI, SWI	↓ T_1_ signal in SNpc and habenular nuclei. ↓ T_2_* in STN. ↑ QSM values in STN. = PDWI values.
Lakhani et al., 2024 [80]	NM-MRI, R_2_*, SWI	↓ SNpc volume. ↓ NM in SNpc.

=, The same as in controls; ↑, increased compared to controls; ↓, reduced compared to controls; AD, axial diffusivity; ASL, arterial spin labeling; CBF, cerebral blood flow; Cho, choline-containing compounds; Cre, creatine; DTI, diffusion tensor imaging; DWI, diffusion-weighted imaging; FA, fractional anisotropy; FC, functional connectivity; FLAIR, T_2_-weighted Fluid-Attenuated Inversion Recovery; fMRI, functional MRI; FSWM, fronto-striatal white matter; IPC, inferior parietal cortex; LC, locus coeruleus; LN, lentiform nucleus; MD, mean diffusivity; MRI, magnetic resonance imaging; MRS, magnetic resonance spectroscopy; NAA, N-acetyl aspartate; NM-MRI, neuromelanin-sensitive imaging; PDWI, proton-density-weighted imaging; pMRI, perfusion MRI; PVS, perivascular space; QSM, quantitative susceptibility mapping; R_2_*, (1/T_2_*)-weighted imaging; RD, radial diffusivity; SMA, supplementary motor area; SN, substantia nigra; SNpc, SN pars compacta; STN, subthalamic nucleus; SWI, susceptibility-weighted imaging; T_1_, T_1_-weighted imaging; T_1_-FSE, T_1_-weighted fast spin-echo; T_2_*, T_2_*-weighted imaging; T_2_, T_2_-weighted imaging.

### 4.2. Progressive Supranuclear Palsy

PSP is characterized by significant midbrain atrophy, which is easily visible on sagittal T_1_-weighted MRI scans and serves as a hallmark feature distinguishing PSP from PD. This distinctive atrophy, often described as the “hummingbird” or “penguin” sign, is accompanied by shrinkage in other regions, including the basal ganglia, frontal cortex, insula, and thalamus, which are also associated with abnormal spectroscopy and diffusion imaging findings [29,81]. A recent review summarized structural, diffusion, QMT, and fMRI findings in PSP and MSA [28].

VBM analyses of the entire brain in PSP have demonstrated that lesions predominantly affect the midbrain and result in reduced grey and white matter volumes in the basal ganglia, frontal cortex, insula, and thalamus [82,83]. The midbrain atrophy in PSP is most evident in sagittal images, forming the characteristic hummingbird or penguin silhouette, while in axial images, it is identified by the enlargement of the interpeduncular cistern and the visualization of the peri-mesencephalic cistern [84,85]. VBM techniques have been employed to measure the midbrain-to-pons ratio and the Magnetic Resonance Parkinsonism Index (MRPI), achieving high diagnostic accuracy in distinguishing PSP patients from both PD and MSA patients, with accuracies of 98% and 91%, respectively [86,87]. The MRPI, which integrates these ratios with measurements of the middle and superior cerebellar peduncles, further enhances diagnostic accuracy [88] and is useful for longitudinal evaluation of disease progression [89].

Signal changes in PSP include hyperintensities in the midbrain and hypointensities in the posterolateral putamen on T_2_-weighted images. However, these findings are not reliable markers for differentiating PSP from PD [85,90]. Recent studies have demonstrated that the putamen ratio of T_1_/T_2_ images is statistically larger in PSP compared to PD patients, providing a potential diagnostic marker [91]. Functional MRI studies have revealed reduced FC in PSP patients in prefrontal–parietal cortical connections. Longitudinally, PSP patients exhibit declining FC between subcortical–brainstem and parietal modules, along with atrophy in the midbrain, pallidum, and perirolandic cortex [92]. Resting-state FC is higher in PSP patients compared to healthy controls in the default mode network and cerebellar resting-state networks (RS-networks), while reduced FC is observed between the cerebellum and the insula, as well as the lateral visual and auditory cortices [93].

DWI and DTI studies have shown increased diffusivity or reduced FA in the basal ganglia, midbrain, superior cerebellar peduncle, and precentral white matter in PSP [53,56]. Free-water imaging has revealed longitudinal declines in descending sensorimotor tracts in MSA and PSP compared to PD, with FA reductions observed in gray matter regions such as the SN, putamen, subthalamic nucleus, red nucleus, and pedunculopontine nucleus in PSP [94]. MRS studies have reported reduced NAA/Cre ratios in the lentiform nucleus of PSP patients, reflecting neuronal loss in this region [95,96]. The most effective biomarkers for distinguishing PSP from healthy controls include neuromelanin-based SN volume, midbrain FA, midbrain, globus pallidus, and putamen volumes, as well as the FA of the locus coeruleus [97]. SWI measurements in the red nucleus and the medial part of the SN in PSP, and in the putamen in MSA, have been successfully used as biomarkers to differentiate these conditions from PD [98].

### 4.3. Multiple System Atrophy

In contrast to PSP and PD, MSA predominantly affects the cerebellum and pons, resulting in notable atrophy in these regions. MRI indices, such as an increased midbrain-to-pons ratio, are useful for distinguishing MSA from PSP or PD. Similar to PSP, cerebral atrophy in MSA extends to the basal ganglia, frontal cortex, insula, and thalamus, with T_2_-weighted images revealing specific changes, including a lateral hyperintense halo and dorsolateral signal attenuation in the putamen, as well as the characteristic “hot cross bun” sign in the pons [29,81].

MSA is a neurodegenerative disorder that affects various brain regions, including the basal ganglia, brainstem, cerebellum, and spinal cord [99,100]. Pathologically, it is characterized by α-synuclein inclusions in oligodendrocytes, which disrupt the function of affected brain areas [101]. Clinically, MSA presents with a combination of parkinsonism, cerebellar ataxia, and autonomic failure. Depending on the dominant symptoms, it is classified into two variants: MSA-P (parkinsonism) and MSA-C (cerebellar) [102,103].

A combination of T_1_, T_2_, T_2_*, DTI, and VBM measurements applied to the entire brain serves as biomarkers to distinguish MSA from PD patients [104]. Additionally, the posterior hypothalamus volume is smaller in MSA patients compared to both controls and PD [105]. Conventional MRI, when combined with VBM, reveals atrophy in multiple brain regions, particularly the cerebellum, pons, and middle cerebellar peduncles, with MSA-C exhibiting more pronounced atrophy in these regions compared to MSA-P [106]. T_2_-weighted images of MSA reveal a distinctive hyperintense halo in the putamen and abnormal signal attenuation in the dentate nucleus [107]. VBM studies have demonstrated that the iVAC parameter (individual VBM adjusted for covariates) is effective in distinguishing MSA from PD patients by measuring atrophy in the pons and putamen [108].

FC studies have identified reduced local efficiency, weighted degree, and betweenness centrality (BC) in the cerebellum and several cortical regions of MSA patients, while BC was increased in the left dorsolateral prefrontal cortex and left cerebellum [109]. Another study reported significant FC differences between the central autonomic network and brain areas such as the putamen or cerebellum, as well as between the sensorimotor control and limbic networks. Furthermore, changes in FC were positively correlated with the severity of MSA [110].

DWI and DTI studies have shown increased diffusivity and reduced FA in the pons, cerebellar peduncles, and putamen, with diffusion measurements of these areas distinguishing MSA from PSP with high sensitivity and specificity [53,111]. Free-water imaging has revealed longitudinal declines in descending sensorimotor tracts in MSA, with FA reductions observed in gray matter regions such as the putamen [94]. Reduced streamline counts were found in MSA compared to controls for connections involving the putamen, ventral diencephalon, pallidum, thalamus, and cerebellum [112]. MRS studies have confirmed reduced NAA/Cre ratios in the lentiform nucleus and pons, indicative of neuronal loss [113]. SWI has been used to identify hypointensity in the putamen of MSA patients, providing a useful biomarker for distinguishing MSA from other movement disorders, including PD, LBD, and other related conditions [114].

### 4.4. Corticobasal Degeneration

CBD presents significant diagnostic challenges due to its overlapping symptoms with other movement disorders. Conventional MRI often reveals asymmetric cortical atrophy, particularly in the parietal and frontal lobes, which correlates with clinical features such as apraxia, rigidity, and myoclonus. VBM studies have demonstrated cortical thinning, subcortical volume loss, and fiber tract degeneration in the hemisphere contralateral to the more affected limb [115]. Reduced gray matter density has been observed in the basal ganglia, insula, thalamus, premotor cortices, and SMA, suggesting that CBD pathology extends beyond the cortex to subcortical regions [116]. A review by Constantinides et al. [117] summarizes findings from various imaging techniques, including MRI, which have identified asymmetrical cortical atrophy in the perirolandic region (anterior and posterior central gyri), posterior frontal, and parietal lobes, contralaterally to the clinically more severely affected side.

NM-MRI, though less extensively studied in CBD, may also reveal degeneration in the SN, with lower contrast observed in CBD patients compared to controls in this region, as well as between hemispheres in the substantia nigra and locus coeruleus. Differentiating CBD from other tauopathies, such as PSP and FTD, remains challenging due to overlapping MRI findings [118].

FC measurements have shown reduced thalamic–cortical, thalamic–subcortical, and cerebellar connectivity in CBD patients. Additionally, CBD patients exhibit reduced FC between the lateral visual and auditory RS-networks, and increased FC in the salience and executive-control RS-networks compared to controls [93,115].

DWI and DTI have revealed white matter abnormalities in corticospinal and corpus callosum tracts, which are frequently affected in CBD. These techniques are valuable for identifying motor and cognitive dysfunctions associated with the disease [119]. Restricted diffusion values have also been observed in the parieto-occipital gyri bilaterally in CBD patients [120].

### 4.5. Lewy Body Dementia

LBD is a common cause of dementia in the elderly, ranking second only to Alzheimer’s disease (AD) in prevalence. It shares clinical features with both AD and PD, complicating its diagnosis. LBD is characterized by fluctuating cognition, recurrent visual hallucinations, RBD, and spontaneous parkinsonism [121].

Conventional MRI findings in LBD include diffuse cortical atrophy, particularly in posterior brain regions such as the occipital lobe, as well as in the occipital, temporal, right frontal, and left parietal lobes, putamen, hippocampus, parahippocampal region, anterior cingulate gyrus, nucleus accumbens, and thalamic nuclei [50]. This pattern of atrophy contrasts with AD, where atrophy is more prominent in the hippocampus and temporal lobes [122]. Borroni et al. [123] identified structural and functional differences between LBD patients and healthy controls, demonstrating atrophy in parietal and occipital structures, as well as reduced FC in posterior regions. Recently, larger atrophy values have been correlated with specific psychological phenotypes, such as euphoric/hypersexual behavior, and positive correlations have been found with the right thalamus, basal forebrain structures, mesial temporal lobes, the striatum, orbitofrontal cortex, anterior cingulate cortex, inferior, superior, and middle temporal lobe, and the inferior frontal gyrus [124].

Advanced imaging techniques, such as NM-MRI, are essential for detecting degeneration in the SN and locus coeruleus, key regions affected by α-synuclein pathology in LBD. The loss of neuromelanin-containing neurons in these areas correlates with the severity of both parkinsonism and cognitive decline [80]. A recent meta-analysis revealed that DLB patients exhibited FC differences in the bilateral inferior parietal lobule and right lingual gyrus compared to healthy controls. The frontal–parietal, salience, and visual RS-networks were abnormally coactivated in DLB, while the default mode network remained normally coactivated compared to AD [121].

DTI has revealed microstructural damage in white matter tracts connecting the thalamus, occipital cortex, and basal ganglia, aiding in the differentiation of LBD from AD and other dementias. Reduced FA values have been observed in the corpus callosum, caudate nucleus, pericallosal regions, inferior longitudinal fasciculus, precuneus, amygdala, frontal, parietal, and occipital white matter compared to controls [81]. MRS studies have reported a reduced NAA/Cre ratio in the hippocampus when comparing PD with LBD and healthy controls [125].

### 4.6. Frontotemporal Dementia

FTD is a group of neurodegenerative diseases affecting the frontal and temporal lobes, as well as subcortical structures such as the thalamus. FTD represents the clinical syndrome, while frontotemporal lobar degeneration refers to the underlying neuropathological changes [126]. Three distinct clinical variants are recognized: behavioral variant (bvFTD), characterized by behavioral and personality changes with frontal cortical degeneration; semantic dementia, marked by progressive loss of word and object knowledge due to anterior temporal neuronal loss; and primary progressive aphasia (PPA), featuring effortful speech, grammar loss, and motor speech deficits associated with left perisylvian atrophy. FTD overlaps clinically and pathologically with atypical parkinsonian disorders (e.g., CBD, PSP) and amyotrophic lateral sclerosis, contributing to significant diagnostic complexity [127].

MRI often reveals frontal lobe atrophy in bvFTD, particularly affecting regions such as the anterior cingulate cortex and insula [128]. In PPA, atrophy is more localized to the left perisylvian cortex, including the temporal and inferior frontal regions [129]. rsfMRI studies have demonstrated disruptions in functional connectivity within the default mode and salience networks, which are critical for social cognition and behavior. These disruptions are more pronounced in bvFTD and correlate with symptoms such as disinhibition and apathy [126,130].

DTI and DWI further elucidate the network disruptions underlying FTD. DTI has shown white matter loss in frontotemporal connections, including the uncinate fasciculus and corpus callosum [131]. This technique is also useful for differentiating between different FTD subtypes [132]. DWI has identified abnormalities in several tracts, including the bilateral uncinate fasciculus, frontal callosum, anterior thalamic radiations, cingulum bundles, and left superior longitudinal fasciculus [133]. MRS studies have found a reduced NAA/Cre ratio in the posterior cingulate cortex for FTD patients [134].

Future advances in tau-sensitive MRI may help differentiate FTD from other tauopathies, including Alzheimer’s disease and CBD [135].

**Table 3 brainsci-15-00079-t003:** PD-based disorders (PSP, CBD, MSA, LBD, FTD). A collection of key references on using MR techniques for diagnosing and monitoring PD-related disorders, with an emphasis on recent studies.

Dis.	Study	MR Techniques	Main Findings
PSP	Tedeschi et al., 1997 [96]	T_1_, T_2_, MRS	↓ NAA/Cre in brainstem, centrum semiovale, frontal, and precentral cortex. ↓ NAA/Cho in lentiform nucleus.
	Brenneis et al., 2004 [83]	T_1_ (VBM)	↓ Grey matter volume in prefrontal cortex (medial and left lateral middle frontal gyrus), insula (frontal opercula), SMA and left medio-temporal area. ↓ White matter volume in frontotemporal and mesencephalon regions.
	Nicoletti et al., 2006 [53]	T_1_, T_2_, PDWI, DWI	Atrophy of the midbrain and putamen, third ventricle dilation, ↑ T_2_ in periaqueductal. ↑ ADC in putamen, caudate, globus pallidus, thalamus.
	Seppi et al., 2010 [29]	T_1_, T_2_, T_2_*, PDWI, VBM, DWI, DTI, QMT	Atrophy in midbrain (“penguin silhouette” or “hummingbird” sign), tegmental area, superior cerebellar peduncle, frontal and temporal lobes. Enlargement of third ventricle. ↑ T_2_ in midbrain and inferior olives. ↓ Volume in whole brain, striatum, brainstem (midbrain), frontal lobe. ↑ Diffusivity in superior cerebellar peduncle and ↓ FA in midbrain. ↓ MTR in SN.
	Bharti et al., 2017 [93]	rsfMRI	↑ FC in default mode and cerebellum networks. ↓ FC between cerebellum and insula. ↓ FC between lateral visual and auditory networks.
	Heim et al., 2021 [86]	T_1_	MRPI and M/P ratio showed high diagnostic accuracy in distinguishing PSP from MSA.
	Mazzucchi et al., 2022 [98]	T_1_, SWI	↑ MRPI and ↑ P/M. MRPI and P/M showed high accuracy in distinguishing PSP from PD but not for MSA. QSM values in red nucleus, putamen, and SN differentiate PD, PSP, and MSA.
	Kawazoe et al., 2024 [118]	NM-MRI	↓ NMR in SN.
MSA	Watanabe et al., 2004 [113]	MRS	↓ NAA/Cre in putamen and pons.
	Nicoletti et al., 2006 [53]	T_1_, T_2_, PDWI, DWI	↑ ADC in middle cerebellar peduncle, putamen and caudate.
	Minnerop et al., 2007 [106]	VBM, VBR	↓ Volumes in cerebellum, right thalamus, both putamina, cortical regions. ↓ R_2_ in cerebellum, pontine brainstem, cortical regions.
	Seppi et al., 2010 [29]	T_1_, T_2_, T_2_*, PDWI, VBM, DWI, DTI, QMT	↓ T_2_ in putamen. Atrophy in putamen, pons (“hot-cross bun” sign), cerebellum. ↓ Volume in striatum, brainstem, cerebellum. ↑ ADC and ↓ FA in putamen. ↓ MTR in putamen.
	Péran et al., 2018 [104]	T_1_, T_2_, T_2_* and PDWI, DTI and VBM	T_1_, T_2_, T_2_*, DTI, and VBM measurements combined are biomarkers to distinguish MSA from PD patients. Atrophy in putamen, pons, cerebellum. ↑ R_2_* in putamen and cerebellum. ↑ MD and ↓ FA in cerebellum, brain stem, superior corona radiata, corpus callosum, external and internal capsules.
	Lyu et al., 2023 [110]	rsfMRI	Altered FC between central autonomic network and putamen, cerebellum, sensorimotor control, and limbic networks. ↓ FC in insula, putamen, and cerebellum. ↑ FC in superior frontal gyrus, posterior middle temporal gyrus, hippocampus. FC alterations correlate with disease severity.
	Kawazoe et al., 2024 [118]	NM-MRI	↓ NMR in SN and locus coeruleus.
CBD	Tedeschi et al., 1997 [96]	T_1_, T_2_, MRS	↓ NAA/Cre in centrum semiovale. ↓ NAA/Cho in lentiform nucleus and parietal cortex.
	Seppi et al., 2010 [29]	T_1_, T_2_, T_2_*, PDWI, VBM, DWI, DTI, QMT	Atrophy in frontoparietal cortex and corpus callosum. ↑ T_2_ in motor cortex, subcortical white matter. ↓ T_2_ in putamen. ↑ Diffusivity in middle cerebellar peduncle.
	Bharti et al., 2017 [93]	rsfMRI	↑ FC in default mode, cerebellum, sensorimotor, executive-control, and insula networks. ↓ FC between the lateral visual and auditory networks. ↑ FC between salience and executive-control networks.
	Constantinides et al., 2019 [117]	T_1_, T_2_ T_2_*, PDWI, VBM, DWI, DTI, rsfMRI	Asymmetrical cortical atrophy in the perirolandic region, posterior frontal, and parietal lobes, midbrain, corpus callosum, basal ganglia. ↑ T_2_ in basal ganglia. ↑ MD in central, middle, superior, and inferior frontal gyri. ↓ FA and ↑ ADC in corticospinal tract and posterior corpus callosum. ↓ Cortical thickness in prefrontal cortex, precentral gyrus, SMA, insula, and temporal pole. ↓ Volume in putamen, hippocampus, accumbens, corpus callosum. Altered FC in thalamic and cerebellar dentate nucleus networks.
	Kawazoe et al., 2024 [118]	NM-MRI	↓ NMR in SN.
LBD	Bozzali et al., 2005 [136]	DWI, DTI	↓ Grey matter volume. ↑ MD and ↓ FA in frontal, parietal, occipital, corpus callosum, pericallosal areas, caudate nucleus, putamen.
	Xuan et al., 2008 [125]	MRS	↓ NAA/Cre in hippocampus.
	Pizzi et al., 2016 [122]	T_1_	↓ Volume in hippocampus. ↓ Thickness of perirhinal cortex and parahippocampus.
	Saeed et al., 2020 [50]	T_1_, T_2_, T_2_*, R_2_*, PDWI, NM-MRI, FLAIR, SWI, DWI, DTI, fMRI, MRS	↓ Volumes in occipital, temporal, right frontal and left parietal lobes, putamen, hippocampus, parahippocampal region, anterior cingulate gyrus, nucleus accumbens, and the thalamic nuclei. ↑ MD and ↓ FA in corpus callosum, pericallosal regions, caudate nucleus, amygdala, inferior longitudinal fasciculus, precuneus, frontal, parietal, and occipital lobe. ↓ NAA/Cr in hippocampus.
	Ma et al., 2022 [137]	rsfMRI	↑ Node degree (a measure of connectivity) in frontal and parietal lobes. ↓ Node degree in SMN and auditory network. ↑ Rich club nodes (highly interconnected regions) in the frontoparietal network. ↓ Rich club nodes in the SMN.
FTD	Mihara et al., 2006 [134]	MRS	↓ NAA/Cre, ↓ Cho/Cre and ↑ MI/Cre in frontal white matter. ↓ NAA/Cre in posterior cingulate cortex.
	Rabinovici et al., 2007 [129]	VBM	Atrophy in medial prefrontal and medial temporal cortex, insula, hippocampus, amygdala, anterior cingulate, subcallosal gyrus, striatum.
	Zhang et al., 2009 [131]	DWI, DTI	↓ FA, ↑ RD and ↑ AD in anterior corpus callosum, anterior and descending cingulum, uncinate tracts.
	Daianu et al., 2016 [133]	DWI, DTI	↑ MD, ↑ RD, ↑ AD and ↓ FA in uncinate fasciculus, frontal segment of the corpus callosum, anterior thalamic radiation fibers, cingulum bundles, superior longitudinal and inferior fronto-occipital fasciculus.
	Ducharme et al., 2020 [128]	T_1_, T_2_, VBM, DTI	↓ Volume in frontal and anterior temporal areas.
	Ferreira et al., 2022 [126]	rsfMRI	↓ FC within the salience network (dorsal anterior cingulate cortex, anterior insula). ↓ Overall connectivity and ↓ network efficiency. Altered FC within default mode network. ↓ FC within frontoparietal attentional, executive, dorsal attentional networks.

↑, increased compared to controls; ↓, reduced compared to controls; AD, axial diffusivity; ADC, apparent diffusion coefficient; CBD, corticobasal degeneration; Cho, choline; Cre, creatine; DTI, diffusion tensor imaging; DWI, diffusion-weighted imaging; FA, fractional anisotropy; FC, functional connectivity; fMRI, functional MRI; FTD, frontotemporal dementia; LBD, Lewy body dementia; M/P, midbrain-to-pons ratio; MD, mean diffusivity; MI, myoinositol; MRI, magnetic resonance imaging; MRPI, magnetic resonance parkinsonism index; MSA, multiple system atrophy; MTR, magnetization transfer ratio; NAA, N-acetyl aspartate; PD, Parkinson’s disease; PDWI, proton-density-weighted imaging; PSP, progressive supranuclear palsy; QMT, quantitative magnetization transfer; QSM, quantitative susceptibility mapping; R_2_*, (1/T_2_*)-weighted imaging; RD, radial diffusivity; rsfMRI, resting-state fMRI; SMA, supplementary motor area; SMN, sensori/somatomotor network; SN, substantia nigra; SWI, susceptibility-weighted imaging; T_1_, T_1_-weighted imaging; T_2_*, T_2_*-weighted imaging; T_2_, T_2_-weighted imaging; VBM, voxel-based morphometry; VBR, voxel-based relaxometry.

### 4.7. Huntington’s Disease

HD is a neurodegenerative disorder characterized by early and severe atrophy of the striatum, detectable even in presymptomatic gene carriers, sometimes more than 20 years before motor symptoms manifest. HD is caused by an expanded CAG triplet repeat in the HTT gene, leading to the production of abnormal huntingtin protein. It is inherited in an autosomal dominant manner, with a prevalence of 4–10 cases per 100,000 people in Western populations [138,139]. The disease is marked by progressive motor impairments, cognitive decline, and psychiatric disturbances. Striatal volume correlates negatively with motor and cognitive functions and the length of CAG triplet repeats. Grey matter loss typically begins in the putamen and, to a lesser extent, the caudate nucleus. Over time, this atrophy extends to the cingulate, precentral, and prefrontal cortices, eventually affecting the occipital, parietal, and temporal cortices. MRI can detect striatal atrophy in gene carriers well before motor symptoms appear, with striatal volume reductions correlating with clinical and genetic factors [140,141,142]. Diffusion changes have also been documented in the striatum and white matter, including the corpus callosum and cortico-striatal fibers [138,139,143].

Structural imaging, VBM, cortical thickness, and brain volume analyses using MRI techniques have been thoroughly reviewed by Wilson et al. [144], with results corroborated by other VBM studies [145]. Initial findings showed a reduction in striatal, caudate, and putamen volumes. Premotor, SMA, and sensorimotor cortices have also been found to have reduced cortical thickness in HD patients [146]. Atrophy in the putamen, striatum, thalamus, and frontal lobes is associated with reduced cognitive and emotional testing performance [147]. These atrophies appear at early stages of the disease and change over time, making them ideal biomarkers for studying disease progression [148,149].

In NM-MRI studies, reduced SN area was found in HD patients, along with a decrease in the signal ratio of SN/crus cerebri [150]. SWI studies revealed higher susceptibility values in the caudate nucleus, globus pallidus, and putamen due to high iron levels, while reduced SWI signals were observed in the SN and hippocampus. Susceptibility values in the caudate nucleus and putamen were negatively correlated with their respective volumes [151].

Diffusion MRI techniques have revealed changes in the striatum and white matter tracts, including reduced cortico-striatal connectivity [152]. Damage to the corpus callosum in HD patients has been correlated with cognitive measurements [143]. A review by Liu et al. [153] reported increased FA in the globus pallidus, putamen, and caudate in HD patients, but reduced FA in the corpus callosum. MD increased in the thalamus, putamen, and caudate, while radial and axial diffusivity increased in the caudate compared to controls.

Functional MRI studies have shown reduced FC in HD patients, particularly in networks such as the visual, executive control, and default mode networks [154,155]. Reduced amplitude of low-frequency fluctuations activations were found in the putamen and inferior temporal gyrus, correlating with performance on the Stroop test [156]. Connectivity between distinct networks was reduced, indicating a loss of long-range functional coupling. Conversely, increased FC was observed in subcortical regions such as the caudate nucleus, thalamus, putamen, cerebellum, and large parts of the frontal cortex, including motor areas [157]. A graph theory study on HD patients demonstrated progressive reductions in network interconnectivity, loss of centrality, and clustering parameters, particularly in the sensorimotor network, although these parameters showed high variability and were less effective for longitudinal studies [158].

QMT techniques have been reviewed by Tambasco et al. [159,160,161], who reported reduced magnetization transfer ratio in the brain as a whole, subcortical gray matter, and putamen, all of which were highly correlated with disease duration.

Perfusion MRI studies have been limited in recent years, with few reports available [162]. Wolf et al. [163] found reduced CBF in pre-HD patients compared to controls in the putamen and medial and lateral prefrontal regions, but increased CBF in the hippocampus and precuneus. Chen et al. [164] reported reductions in CBF in the paracentral, sensorimotor, medial occipital, inferior temporal, and lateral occipital regions, insula, postcentral gyrus, and other subcortical regions. In both studies, CBF was correlated with cognitive test performance, indicating its potential as a biomarker for disease progression.

MRS studies have found reduced levels of NAA, Cre, and glutamate in the caudate nucleus and putamen in HD patients compared to controls [165]. A more recent study corroborated these findings and further identified reduced myoinositol (MI) in the putamen, with MI and creatine concentrations negatively correlated with caudate volume [166]. Jing et al. [167] confirmed these findings in the basal ganglia and suggested their potential as future biomarkers for the disease.

### 4.8. Dystonia

Dystonia is characterized by involuntary muscle contractions leading to abnormal postures and movements. Dystonia can be classified as primary or secondary, with secondary dystonia often arising from brain lesions in various regions, including the basal ganglia, thalamus, cerebellum, brainstem, and spinal cord. Lesion location influences the type of dystonia. For example, basal ganglia lesions are often linked to hemidystonia, while thalamic lesions may result in hand dystonia and axial dystonia with cerebellar or brainstem lesions. In craniofacial dystonia, lesions have been reported in the brainstem, cerebellum, and spinal cord [168,169,170]. Conventional MRI often appears normal in primary dystonia, but newer techniques, such as VBM and DWI, have identified structural abnormalities in the sensorimotor network, including the primary motor cortex, basal ganglia, and cerebellum [170].

In a review by MacIver et al. [171], changes in structural, VBM, and DTI studies in dystonia were presented. In primary dystonia, MRI typically shows no significant structural abnormalities. However, spinal cord lesions have been associated with specific types of cervical dystonia, such as torticollis or laterocollis [172]. Waller et al. [173] used T_2_-FLAIR techniques to show hyperintensities in the posterolateral putamen of dystonia patients, which were thought to be associated with leukodystrophy. Egger et al. [174] found increased gray matter volume bilaterally in the nucleus accumbens, globus pallidus internus, prefrontal cortex, and left inferior parietal lobe in dystonia patients compared to healthy controls. In contrast, MacIver et al. [171,175] reported no differences in regional volumes (prefrontal, sensorimotor, non-frontal, thalamic, striatal, or cerebellar) or cortical thickness (prefrontal, sensorimotor, and non-frontal) between healthy controls and dystonia patients using detailed anatomical T_1_-imaging.

fMRI studies, as reviewed by Simonyan et al. [176], have shown reduced FC in the sensorimotor and inferior parietal cortices in dystonia. Decreased connectivity in the primary somatosensory cortex was accompanied by increased FC in the putamen and decoupling of the dorsal premotor cortex from the parietal cortex. Reduced FC and activity were also observed in preparatory cortical regions and the basal ganglia. Additionally, reduced thalamo-cortical connectivity and decreased connectivity within the distal thalamo-cortical segment of the cerebello-thalamo-cortical tract were noted.

DTI has identified increased putamen volume and structural abnormalities in the basal ganglia, cortex, thalamus, and cerebellum. These imaging abnormalities support the growing view that different types of dystonia affect distinct brain regions [111,152,174,177,178]. MacIver et al. [171] confirmed these findings, reporting abnormalities in the brainstem, cerebellum, basal ganglia, and sensorimotor cortex, as well as in their interconnecting white matter pathways. DTI studies have shown abnormalities in critical white matter pathways, including the cerebellothalamic and thalamocortical tracts [179,180,181].

Secondary dystonias are often linked to lesions in multiple brain regions. Lesions in the basal ganglia are most associated with hemidystonia, while thalamic lesions can cause hand dystonia due to the disruption of motor pathways. Axial dystonias, such as cervical dystonia, are frequently associated with cerebellar or brainstem lesions [172,182].

A small number of studies have investigated perfusion MRI in dystonia. For a patient with hand dystonia, increased perfusion in the inferior prefrontal cortex was negatively correlated with disease duration [183]. Yang et al. [184] found increased CBF in the middle frontal gyrus and bilateral precentral gyrus in dystonia patients. No changes in perfusion were observed in the globus pallidus of dystonia patients [185].

Limited work has been conducted on MRS in dystonia. Some studies have shown lactate peaks in the lateral ventricles and cerebral cortex [186]. In a study comparing the motor cortex and lentiform nucleus of dystonic patients and healthy controls, no changes were found in GABA, glutamate plus glutamine, and NAA levels before repetitive transcranial magnetic stimulation treatment [187].

**Table 4 brainsci-15-00079-t004:** Huntington’s disease and dystonia. A collection of what authors consider the main references on MR techniques for diagnosing and monitoring HD and DTN, focused on recent years.

Dis.	Study	MR Techniques	Main Findings
HD	Aylward et al., 2011 [148]	T_1_, T_2_, T_2_*, PDWI, VBM	↓ Volume in striatum (caudate, putamen), globus pallidus, thalamus and frontal lobes.
	Chen et al., 2012 [164]	pMRI	↓ CBF in the cortex (paracentral, sensorimotor, medial occipital areas, inferior temporal and lateral occipital regions, insula and postcentral gyrus) and subcortical regions (caudate and putamen). ↑ CBF in pallidum. ↓ Volume in caudate, putamen, pallidum, thalamus, amygdala, hippocampus.
	Dumas et al., 2013 [155]	rsfMRI	↓ FC within left middle frontal and pre-central gyrus, default mode and executive control networks. ↓ FC between right post-central gyrus, and cingulate gyrus with medial visual network.
	Werner et al., 2013 [157]	rsfMRI, VBM	Grey matter atrophy in striatum, insula, parietal operculum, middle temporal gyrus, left inferior parietal cortex, premotor cortex, and sensorimotor cortex. ↑ FC in thalamus, striatum, prefrontal, premotor, and parietal cortex. ↓ FC between inter-resting-state networks.
	Tambasco et al., 2015 [161]	QMT	↓ MTR in whole brain, subcortical gray matter. ↑ MTR in putamen (premanifest disease).
	van Bergen et al., 2016 [151]	SWI	↑ QSM values in caudate nucleus, putamen, globus pallidus. ↓ QSM values in SN, hippocampus. ↓ Volume and ↑ effective relaxation in caudate nucleus and putamen.
	Gregory et al., 2018 [154]	fMRI (rsfMRI, tbfMRI)	↑ Activation in SMA and superior parietal regions, dorsolateral prefrontal cortex. ↓ Activation in rostral SMA, inferior frontal gyrus, anterior insula, and striatum. ↓ FC between medial prefrontal cortex and left premotor region. ↓ FC within medial visual, dorsal attention, and executive function networks. ↑ FC within default mode and motor networks.
	Wilson et al., 2018 [144]	T_1_, T_2_, T_2_*, PDWI, VBM, SWI, DTI, DWI, QMT	↓ Volume in striatum (caudate, putamen), frontal lobe, frontal white matter. ↑ FA in striatum (putamen, globus pallidus). ↓ FA in corpus callosum, frontal, parietal, and occipital white matter, and surrounding the striatum, thalamus, and corpus callosum. ↑ RD and ↑ AD in corpus callosum, frontal tracts, thalamic tracts, and white matter surrounding the striatum. ↓ MTR in cortical grey and white matter. ↑ MTR in putamen. ↑ [Fe] in basal ganglia (caudate, putamen, globus pallidus). ↓ [Fe] in frontal lobe white matter.
	Leitao et al., 2020 [150]	NM-MRI	↓ Area in SN. ↓ SN to midbrain area ratio. ↓ NM contrast SN/crus cerebri ratio.
	Lowe et al., 2022 [166]	MRS	↓ Volume in caudate. ↓ Cre and ↓ NAA. Cre and MI correlated with caudate volume.
DTN	Egger et al., 2007 [174]	VBM	↑ Volume bilaterally in globus pallidus internus, nucleus accumbens, orbitofontal cortex, medial frontal gyrus, left inferior parietal lobe. = Grey matter volume
	Simonyan et al., 2008 [179]	DTI, DWI	(Laryngeal dystonia). ↓ FA in right genu of the internal capsule. ↑ Water diffusivity in white matter along the corticobulbar/corticospinal tract. ↑ Water diffusivity in lentiform nucleus, ventral thalamus, cerebellar white and gray matter.
	Marjanska et al., 2013 [187]	MRS	= Concentration of NAA, Glx, and GABA in motor cortex, lentiform nucleus, and occipital region. ↓ NAA, ↓ GABA, and ↓ Glx after 5 Hz rTMS in motor cortex. Asymmetry of NAA and Glx in lentiform nucleus.
	Simonyan et al., 2018 [176]	T_1_, VBM, DWI, DTI, rsfMRI	↑ Volume in putamen. ↓ Axonal integrity and ↑ water diffusivity in cortical and subcortical structures along the cortico-striato-pallido-thalamic and cerebello-thalamo-cortical pathways. Structural alterations in frontoparietal, supplementary motor, and primary sensorimotor areas, caudate nucleus, globus pallidus, thalamus, and cerebellum. ↓ FC within primary somatosensory region. ↑ FC in the putamen. ↓ FC between dorsal premotor cortex and parietal cortex. ↓ FC in sensorimotor and inferior parietal cortices. ↓ FC and activity of the preparatory cortical regions and basal ganglia. Dysfunctional striato-thalamo-cortical and cerebellar-thalamo-cortical pathways.
	MacIver et al., 2022 [171]	T_1_, T_2_, T_2_*, PDWI, VBM, DTI, QMT	= Volume in prefrontal, sensorimotor, non-frontal cortex, thalamus, striatum, cerebellum. ↓ FA and ↑ MD in brainstem, cerebellum, basal ganglia, thalamus and sensorimotor cortex. ↑ R_2_* in globus pallidus. ↓ Cerebellothalamic fibers
	Waller et al., 2022 [173]	T_2_-FLAIR	(*EIF2AK2*-related dystonia). ↑ T_2_ in putamen, frontal and posterior periventricular white matter
	Yang et al., 2023 [184]	pMRI, fMRI	(Meige syndrome). ↑ Whole gray matter CBF–FCS coupling. ↑ CBF in the middle frontal gyrus and precentral gyrus
	Maciver et al., 2024 [175]	T_1_, DWI, DTI, DKI, TG, NODDI	= Volume in prefrontal, sensorimotor, non-frontal, thalamic striatal, cerebellar regions = Cortical thickness in prefrontal, sensorimotor and non-frontal areas. ↑ RK in striatum. = FA, MD, MK, AK, RK in white matter. ↑ FA in mid-right superior cerebellar peduncle. ↑ FA and ↓ ODI in anterior thalamic radiations. ↓ ODI in striatopremotor tracts.

=, The same as in controls; ↑, increased compared to controls; ↓, reduced compared to controls; AD, axial diffusivity; ADC, apparent diffusion coefficient; AK, axial kurtosis; CBD, corticobasal degeneration; CBF, cerebral blood flow; Cho, choline; Cre, creatine; DKI, diffusion kurtosis imaging; DTI, diffusion tensor imaging; DTN, Dystonia; DWI, diffusion-weighted imaging; FA, fractional anisotropy; FC, functional connectivity; FCS, functional connectivity strength; fMRI, functional MRI; FTD, frontotemporal dementia; GABA, γ-aminobutyric acid; Glx, glutamate plus glutamine; HD, Huntington’s disease; LBD, Lewy body dementia; M/P, midbrain-to-pons ratio; MD, mean diffusivity; MI, myoinositol; MK, mean kurtosis; MRI, magnetic resonance imaging; MRPI, magnetic resonance parkinsonism index; MSA, multiple system atrophy; MTR, magnetization transfer ratio; NAA, N-acetyl aspartate; NODDI, neurite orientation dispersion and density imaging; ODI, orientation dispersion index; PD, Parkinson’s disease; PDWI, proton-density-weighted imaging; PSP, progressive supranuclear palsy; QMT, quantitative magnetization transfer; QSM, quantitative susceptibility mapping; R_2_*, (1/T_2_*)-weighted imaging; RD, radial diffusivity; RK, radial kurtosis; rsfMRI, resting-state fMRI; rTMS, repetitive transcranial magnetic stimulation; SMA, supplementary motor area; SN, substantia nigra; SWI, susceptibility-weighted imaging; T_1_, T_1_-weighted imaging; T_2_*, T_2_*-weighted imaging; T_2_, T_2_-weighted imaging; tbfMRI, task-based fMRI; TG, tractography; VBM, voxel-based morphometry; VBR, voxel-based relaxometry.

## 5. Discussion

The integration of MRI techniques into the study and management of movement disorders represents a transformative shift in both clinical practice and research. Traditional clinical trials have relied heavily on subjective clinical scales and retrospective analyses, which have limited the ability to track early neurodegenerative processes objectively. However, the advent of advanced MRI methods, including neuromelanin-sensitive imaging, diffusion-weighted imaging, magnetization transfer imaging, and relaxometry, has provided objective biomarkers that offer real-time insights into microstructural and functional changes in the brain [2,64,135,166]. These techniques promise to reshape clinical trial design, patient outcomes, and therapeutic development by enabling more personalized and precise approaches to treatment.

Historically, clinical trials in PD have depended on motor assessments and patient-reported outcomes, which fail to capture early neurodegeneration. MRI techniques, such as NM-MRI and DWI, can monitor the preservation of dopaminergic neurons in the SN long before motor symptoms appear [2]. These imaging biomarkers should become primary or secondary endpoints in neuroprotective trials, allowing more accurate assessment of therapies designed to slow disease progression. Additionally, relaxometry, which tracks iron accumulation in the SN, enhances understanding of PD progression and the effects of interventions [50].

Differentiating between movement disorders such as PSP, MSA, and PD remains challenging due to overlapping clinical features. MRI techniques can help distinguish these conditions by highlighting distinct neuroanatomical profiles. For example, in PSP, volumetric and diffusion-based MRI reveal midbrain atrophy and connectivity disruptions in the superior cerebellar peduncle, while in MSA, pontocerebellar atrophy and iron deposition in the basal ganglia can be identified [94]. These MRI markers should be incorporated into clinical trials to assess the impact of disease-modifying therapies on specific brain regions and to differentiate between disease subtypes more precisely.

CBD poses another diagnostic challenge due to its asymmetrical presentation and overlapping pathology with other movement disorders. DWI and VBM are instrumental in tracking white matter changes in corticospinal and callosal pathways and gray matter degeneration in the parietal and frontal cortices [117]. Clinical trials should leverage these biomarkers for more effective patient stratification and to assess therapeutic outcomes related to motor and cognitive functions. Given the rarity of CBD, multicenter trials with standardized MRI protocols are essential to generate robust, generalizable data.

LBD shares many clinical and pathological characteristics with both PD and Alzheimer’s disease, complicating diagnosis and treatment development. NM-MRI offers a non-invasive tool to detect early neurodegeneration in the SN and locus coeruleus, critical regions involved in LBD [188]. These techniques should be incorporated into clinical trials targeting α-synuclein aggregation and neuroinflammation as early indicators of therapeutic success. Additionally, rsfMRI can reveal disruptions in network connectivity, providing insights into the progression of cognitive and behavioural symptoms [121].

FTD trials have traditionally focused on symptomatic treatments, but MRI offers a more precise understanding of the disease’s neuroanatomical progression. Techniques like cortical thickness measurements, DWI, and FC analyses can track the atrophy and network dysfunction that characterize different FTD subtypes [126,189]. These imaging biomarkers should serve as primary endpoints in trials testing disease-modifying therapies. The heterogeneity of FTD makes it critical to use MRI-based biomarkers for precise patient stratification and to develop personalized therapeutic strategies.

Despite these advances, significant challenges remain in translating these imaging methods into routine clinical practice. Standardizing MRI protocols across centers is essential for ensuring reproducibility and diagnostic consistency in both neuroimaging research and clinical trials. Variability in MRI equipment, acquisition parameters, participant positioning, and analytical pipelines can lead to significant differences in results, even when the same condition or data are studied. This undermines the reliability of findings, complicates the integration and comparison of data across studies, and hinders the development of reliable biomarkers. Initiatives such as the Brain Imaging Data Structure (BIDS) and the harmonized hippocampus project address these issues by providing standardized frameworks for data acquisition, organization, and analysis. By harmonizing protocols, minimizing artifacts like head motion, and ensuring consistent preprocessing methods, researchers can reduce methodological discrepancies, enhance the comparability of results, and improve the robustness and generalizability of findings. These practices are particularly crucial for advancing translational research in psychiatry, neurodegenerative diseases, and movement disorders [4,190]. Combining multiple MRI modalities—such as neuromelanin imaging, DWI, and relaxometry—could enhance diagnostic accuracy. Multimodal imaging has the potential to provide a comprehensive neurodegenerative profile, which is essential for distinguishing between disorders that often present with overlapping clinical features [191].

The integration of artificial intelligence with advanced magnetic resonance imaging techniques has shown remarkable potential in enhancing the diagnosis, evaluation, and monitoring of movement disorders. By leveraging AI-driven algorithms, such as support vector machines, logistic regression, and deep learning models like convolutional neural networks, researchers have achieved high diagnostic accuracy in conditions such as Parkinson’s disease, essential tremor, and Huntington’s disease. Structural MRI, functional MRI, and diffusion tensor imaging have been pivotal in identifying affected brain regions, with structural MRI achieving the highest sensitivity (93%) for detecting regional abnormalities and mixed modalities demonstrating the highest specificity (90%) [192]. These advanced tools not only improve diagnostic precision but also enable the tracking of disease progression, evaluation of treatment efficacy, and identification of imaging biomarkers. Moreover, AI technologies can process large volumes of complex MRI data with high precision to uncover subtle patterns, predict disease trajectories, and personalize treatment strategies, offering a dynamic and patient-centered approach to care. However, challenges such as data privacy, standardization, and validation across diverse cohorts must be addressed to fully translate these advancements into routine clinical practice, offering a dynamic approach to managing movement disorders and optimizing patient care [193,194].

This review provides a valuable outlook of advanced MRI techniques in movement disorders, yet it is not without limitations. As a narrative review, it lacks the methodological rigor and reproducibility of systematic reviews, as the selection of included studies relies on the authors’ discretion, introducing potential biases in the representation of findings. Moreover, the absence of standardized criteria for study inclusion makes it challenging to validate the conclusions drawn. To enhance future research, adopting systematic review methodologies, including predefined search and inclusion criteria and quantitative synthesis methods like meta-analysis, would improve reproducibility and minimize bias. Additionally, integrating more diverse datasets and exploring emerging technologies such as AI-assisted MRI analysis could further solidify the field’s foundation and broaden its clinical applicability.

## 6. Conclusions

To truly transform clinical trials in movement disorders, advanced MRI must become a central component of trial design. Longitudinal imaging should be standard practice, with baseline scans and regular follow-ups to monitor neuroanatomical changes. Artificial intelligence can further enhance the analysis of complex MRI datasets, allowing the detection of subtle changes that may predict disease progression or therapeutic response earlier than traditional clinical assessments. Multicenter collaborations are also essential to maximize the potential of advanced MRI in clinical trials. Standardized imaging protocols and centralized data analysis will ensure consistency and reproducibility across studies. Moreover, combining MRI with other emerging biomarkers, such as cerebrospinal fluid or blood-based assays for tau or α-synuclein, can improve the sensitivity of these trials to detect early therapeutic effects and refine patient selection criteria.

The future of MRI in movement disorders and neurodegenerative diseases is poised to transform clinical care and research. By integrating these advanced imaging techniques into clinical trials and everyday practice, we stand on the cusp of significant breakthroughs in early diagnosis, disease monitoring, and therapeutic development. These advances will enable earlier interventions, allowing more tailored treatment strategies based on the unique neurodegenerative profiles of individual patients, ultimately improving outcomes and enhancing quality of life.

## Figures and Tables

**Table 1 brainsci-15-00079-t001:** Overview of magnetic resonance imaging techniques for movement disorder research.

Category	Technique	Description
Qualitative MRI	Structural Image Analysis Techniques	
	Proton-density-weighted image (PDWI)	Weighs the signal intensity based on the number of protons, providing information about tissue composition (water and fat content).
	T_1_-weighted	Emphasizes differences in the longitudinal relaxation time, useful for anatomical detail and tissue contrast.
	T_2_-weighted	Highlights differences in the transverse relaxation time, useful for identifying oedema, inflammation, and fluid-filled structures.
	T_2_*-weighted	Sensitive to magnetic field inhomogeneities, useful for detecting microbleeds and iron deposits.
	Adiabatic Techniques	
	T_1ρ_	Measures spin–lattice relaxation in the rotating frame, useful for studying water–protein interactions and assessing brain cell health.
	T_2ρ_	Measures spin–spin relaxation in the rotating frame, sensitive to high iron content, aiding in detecting brain iron buildup.
	Voxel-based morphometry (VBM)	Quantifies brain anatomy differences by analyzing voxel-level tissue density or volume to identify structural abnormalities.
	Susceptibility weighted imaging (SWI)	Enhances contrast from paramagnetic substances (e.g., deoxyhemoglobin, iron, calcifications), useful for detecting microbleeds, veins, and iron deposits.
	Neuromelanin-sensitive imaging (NM-MRI)	Highlights neuromelanin-rich regions, useful for studying dopaminergic and noradrenergic neurons.
	Functional MRI (fMRI)	Measures brain activity by detecting changes in blood oxygenation (BOLD signal).
	Diffusion-weighted imaging (DWI)	Measures the diffusion of water molecules, useful for detecting ischemic stroke and other pathological conditions.
	Diffusion tensor imaging (DTI)	Provides detailed information on the orientation and anisotropy of water diffusion, useful for mapping white matter tracts.
	Magnetic resonance elastography (MRE)	Measures the mechanical properties of tissues, such as stiffness, often used in liver and brain.
Quantitative MRI	Morphometric measurements	Measures cortical thickness and brain volumes to assess structural changes over time, useful for studying brain atrophy
	Quantitative magnetization transfer (QMT)	Provides quantitative data on the exchange of magnetization between different pools of protons, useful for studying tissue microstructure.
	Perfusion MRI (pMRI)	Measures blood flow in the brain, useful for assessing vascular health and brain activity.
	Magnetic resonance spectroscopy (MRS)	Provides information about the chemical composition of tissues by measuring the magnetic resonance spectra, useful for studying metabolic changes.
Other MRI Techniques	Chemical shift	Highlights differences in the chemical environment of protons, useful for distinguishing between different tissues and compounds.
	Artificial intelligence (AI)-assisted MRI	Utilizes machine learning and deep learning algorithms to enhance image acquisition, reconstruction, and analysis. Applications include noise reduction, accelerated imaging, lesion detection, segmentation, and classification for diagnostic purposes. AI models can also predict disease progression and response to therapy using large MRI datasets.

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
