# Peer review of "Advanced Magnetic Resonance Imaging for Early Diagnosis and Monitoring of Movement Disorders"

_brainsci, 2025, doi:10.3390/brainsci15010079_

Round 1

Reviewer 1 Report

Comments and Suggestions for Authors

dear authors

Thank you for choosing this topic for your research because it will indeed impact the treatment of many cases ,however there are some points that need clarification or modification, such as:

1-reference 2-3 not found within the introduction or MRI techniques  ref 2 found later at 3. MRI in Movement Disorders  so please check the Arrangement of  the references

2-no evidence /references for :2.1.1. Structural Image Analysis Techniques. Traditional Imaging with Proton Density, 78 T1, T2 and T2* Contrasts or 2.1.6. Functional MRI or 2.1.7. Diffusion-Weighted Imaging and Diffusion Tensor Imaging or 2.1.8. Magnetic Resonance Elastography

Author Response

Reviewer 1:

Thank you for taking the time to review our manuscript. We sincerely appreciate your thoughtful comments and suggestions, which we believe have significantly improved the quality of our work. All changes have been highlighted in yellow throughout the manuscript for ease of review.

Comment:

Dear authors,
Thank you for choosing this topic for your research because it will indeed impact the treatment of many cases. However, there are some points that need clarification or modification, such as:

  1. Reference 2–3 not found within the introduction or MRI techniques; Ref 2 found later at 3. MRI in Movement Disorders. Please check the arrangement of the references.

Response:

Thank you for identifying the discrepancy in the arrangement of references. We have carefully reviewed and corrected the order of all references throughout the manuscript to ensure consistency and accuracy.

Comment:

  1. No evidence/references for: 2.1.1. Structural Image Analysis Techniques. Traditional Imaging with Proton Density, T1, T2, and T2* Contrasts, or 2.1.6. Functional MRI, or 2.1.7. Diffusion-Weighted Imaging and Diffusion Tensor Imaging, or 2.1.8. Magnetic Resonance Elastography.

Response:

We appreciate this observation and have now thoroughly reviewed the manuscript to address the absence of references in the specified sections. Relevant citations have been added where appropriate to ensure that each technique is adequately supported by evidence from the literature.

Reviewer 2 Report

Comments and Suggestions for Authors

First of all, I would like to thank you for invited to read the document.

The authors have done an excellent job. Each of the comments shared are intended to improve the study.

The comments can be found in the PDF document.

Also, some of the comments on some of the points that need to be reworded in the paper are shared below:

Title

It is suggested to check what type of revision it is and add it to the title.

Abstract

The study is quite interesting because of the subject matter addressed. However, neither in the title nor in the abstract is it clear what the objective/purpose of the study is. This is confusing because it seems to be more of a narrative review.

The abstract is uninformative because it only theorizes elements. It is not clear what knowledge gap it is trying to fill.

Line 78. It is suggested to review if all this conceptual foundation is based on any previous study or research.

Line 146. It is suggested to review if all this conceptual foundation is based on any previous study or research.

Line 171. It is suggested to review if all this conceptual foundation is based on any previous study or research.

Line 215. It is suggested to review if all this conceptual foundation is based on any previous study or research.

Line 246. It is suggested to review if all this conceptual foundation is based on any previous study or research. This would help to strengthen the study. It is suggested to review and rely on background information to strengthen the ideas presented in these subsections of the introduction.

Line 263. It is suggested to review if all this conceptual foundation is based on any previous study or research. This would help to strengthen the study. It is suggested to review and rely on background information to strengthen the ideas presented in these subsections of the introduction.

Material and method (Not reported)

The greatest difficulty presented by the text is that there is no section on material and method that explains the procedures established for conducting the review, eligibility criteria of the studies, background search in different databases, evaluation of the methodological quality of the research. It is also not clear whether the PICOs methodology or some other was used. 

It is also suggested to consider if the present study is a review, if it was registered in any platform (e.g., Prospero, IMPLASY, etc).

Line 441. It is not methodologically clear why these 15 studies are included in this table.

Line 443. It is suggested that the notes to the tables be written in a larger font to explain the acronyms used in the table.

Line 628. It is not methodologically clear why these 31 studies are included in this table.

Line 761. It is not methodologically clear why these 18 studies are included in this table.

Discussion and Conclusions

It is suggested to review the elements to be discussed among the different studies analyzed.

Likewise, separate the conclusions in another section to understand the true scope of this research. 

It is necessary, in turn, to add the limitations of the study and the practical applications or future perspectives of the study.

Finally, I thank the authors for the excellent work and encourage you to review the comments shared.

Author Response

Reviewer 2:

We appreciate the thorough review of our manuscript and the valuable suggestions provided. We have made several changes based on your comments, which we believe have enhanced the clarity and quality of the manuscript. All modifications have been clearly marked in yellow for ease of reference.

Comment:

First of all, I would like to thank you for inviting me to read the document.
The authors have done an excellent job. Each of the comments shared is intended to improve the study.
The comments can be found in the PDF document.
Also, some of the comments on points that need to be reworded in the paper are shared below:

Title:
You should check what type of review it is and add it to the title.

Response:

We have updated the title to indicate the type of review conducted clearly. The revised title now reads: "Advanced MRI in Movement Disorders: A Narrative Review on Its Potential for Early Diagnosis and Monitoring Disease Progression."

Comment:

Abstract:
The study is quite interesting because of the subject matter addressed. However, neither in the title nor in the abstract is it clear what the objective/purpose of the study is. This is confusing because it seems to be more of a narrative review.
The abstract is uninformative because it only theorises elements. It is not clear what knowledge gap it is trying to fill.

Response:

We have revised the abstract to clearly state that this is a narrative review, to provide an updated perspective on the potential of MRI techniques for early diagnosis, disease monitoring, and therapeutic evaluation in movement disorders.

Comment:

Line 78, 146, 171, 215, 246, and 263:
It is suggested to review if all this conceptual foundation is based on any previous study or research. This would help to strengthen the study. It is suggested to review and rely on background information to strengthen the ideas presented in these subsections of the introduction.

Response:

We have carefully reviewed the manuscript and added appropriate references to the sections highlighted. These references now support the conceptual foundation of the diverse MRI techniques discussed, strengthening the ideas presented with relevant prior work as recommended.

Comment:

Material and Methods (Not reported):
The greatest difficulty presented by the text is that there is no section on material and methods that explains the procedures established for conducting the review, eligibility criteria of the studies, background search in different databases, or evaluation of the methodological quality of the research. It is also not clear whether the PICOs methodology or some other was used.
It is also suggested to consider if the present study is a review, if it was registered on any platform (e.g., Prospero, IMPLASY, etc.).

Response:

We have added a Methods section to the manuscript detailing the procedure used for article selection. As this is a narrative review, articles were selected based on the authors' discretion, and no formal guidelines or methodologies for systematic reviews were followed. For this reason, the review was not registered on any platform.

Comment:

Line 441, 443, 628, and 761:
It is not methodologically clear why these 15 studies (Line 441), 31 studies (Line 628), and 18 studies (Line 761) are included in the tables. It is suggested that the notes to the tables be written in a larger font to explain the acronyms used in the table.

Response:

The studies in Tables 2, 3, and 4 were selected at the authors' discretion, based on their representativeness of the main findings and their relevance as key references for MRI techniques in diagnosing and monitoring movement disorders. The selection process was based on consensus among the authors and did not follow a systematic methodology. This clarification has been added to the Methods section. Additionally, the table notes have been revised with a larger font to ensure that all acronyms are clearly explained.

Comment:

Discussion and Conclusions:
It is suggested to review the elements to be discussed among the different studies analysed.
Likewise, separate the conclusions into another section to understand the true scope of this research.
It is necessary, in turn, to add the limitations of the study and the practical applications or future perspectives of the study.

Response:

As suggested, we have separated the Discussion and Conclusion sections. The Discussion section has been expanded to include a critical analysis of the studies reviewed, emphasising the importance of standardising MRI protocols and integrating artificial intelligence into MRI analysis. A paragraph has also been added to outline the study's limitations and suggest future research directions to address these.

Reviewer 3 Report

Comments and Suggestions for Authors

The small review of Dr.Ortega-Robles and colleagues deals with potential application of various advanced magnetic resonance imaging (MRI) in movement disorders. The paper emphasize a necessity to use a combination of objective methods to diagnosis of neurodegenerative diseases and conditions, as well as to monitor them, by using not-yet-developed standardized protocols of various advanced MRIs in everyday clinical practice. Of course, it would be a dream for neurologists. Also, such detailed approach could provide essential knowledge on innate brain compensation processes, which help in tactic short-living tasks but mask the problems and lead to fails in longitudinal strategies.

I believe that this paper will be of interest for practicing neurologists, and also for clinical administrators. If it helps to extend to spectrum of advanced MRI techniques available to ordinary patients, I will be happy.

Author Response

Reviewer 3:

Comment:

The small review of Dr. Ortega-Robles and colleagues deals with potential application of various advanced magnetic resonance imaging (MRI) in movement disorders. The paper emphasise a necessity to use a combination of objective methods to diagnosis of neurodegenerative diseases and conditions, as well as to monitor them, by using not-yet-developed standardised protocols of various advanced MRIs in everyday clinical practice. Of course, it would be a dream for neurologists. Also, such detailed approach could provide essential knowledge on innate brain compensation processes, which help in tactic short-living tasks but mask the problems and lead to fails in longitudinal strategies.

I believe that this paper will be of interest for practicing neurologists, and also for clinical administrators. If it helps to extend to spectrum of advanced MRI techniques available to ordinary patients, I will be happy.

Response:

We sincerely appreciate your thoughtful and encouraging feedback on our manuscript. We are pleased that you consider our review interesting to practising neurologists and clinical administrators.

Your observation regarding the potential of advanced MRI techniques to elucidate innate brain compensation processes is insightful and highly relevant. We agree that understanding these processes could provide crucial knowledge to enhance diagnostic accuracy and refine monitoring strategies in movement disorders. As you highlighted, developing standardised protocols for the clinical implementation of advanced MRI techniques remains a significant challenge. However, achieving this goal would represent a transformative advancement for the field.

We share your aspiration that this work contributes to broadening the accessibility of advanced MRI techniques for a broader patient population. Thank you for recognising the potential impact of our review and for your support in advocating for adopting these methodologies in clinical practice.

Round 2

Reviewer 2 Report

Comments and Suggestions for Authors

First of all, I would like to thank you for invited to read the document.

The authors have done an excellent job.

I have no further comments.